# ProtoCaps: A Fast and Non-Iterative Capsule Network Routing Method

**Miles Everett**                                                                 *m.everett.20@abdn.ac.uk*
*Department of Computing Science*
*University of Aberdeen, UK*

**Mingjun Zhong**                                                                 *mingjun.zhong@abdn.ac.uk*
*Department of Computing Science*
*University of Aberdeen, UK*

**Georgios Leontidis**                                                       *georgios.leontidis@abdn.ac.uk*
*Interdisciplinary Centre for Data and AI*
*Department of Computing Science*
*University of Aberdeen, UK*

**Reviewed on OpenReview:** *https://openreview.net/forum?id=Id1OmlBjcx*

## Abstract

Capsule Networks have emerged as a powerful class of deep learning architectures, known for robust performance with relatively few parameters compared to Convolutional Neural Networks (CNNs). However, their inherent efficiency is often overshadowed by their slow, iterative routing mechanisms which establish connections between Capsule layers, posing computational challenges resulting in an inability to scale. In this paper, we introduce a novel, non-iterative routing mechanism, inspired by trainable prototype clustering. This innovative approach aims to mitigate computational complexity, while retaining, if not enhancing, performance efficacy. Furthermore, we harness a shared Capsule subspace, negating the need to project each lower-level Capsule to each higher-level Capsule, thereby significantly reducing memory requisites during training. Our approach demonstrates superior results compared to the current best non-iterative Capsule Network and tests on the Imagewoof dataset, which is too computationally demanding to handle efficiently by iterative approaches. Our findings underscore the potential of our proposed methodology in enhancing the operational efficiency and performance of Capsule Networks, paving the way for their application in increasingly complex computational scenarios. Code is available at https://github.com/mileseverett/ProtoCaps.

## 1 Introduction

Capsule Networks (Sabour et al., 2017; Hinton et al., 2018; Ribeiro et al., 2020; Hahn et al., 2019; Ribeiro et al., 2022) are a family of Networks designed to overcome the shortcomings of CNNs being unable to handle equivariant translations. Capsule Networks do this by building parse trees of part-whole relationships using groups of individual neurons known as Capsules. While Capsule Networks have been successful in overcoming these shortcomings of CNNs, the core component of the majority of Capsule Networks is iterative, and thus slow and computationally inefficient to scale to images of a realistic size (example can be seen in Figure 1).

Most work in Capsule Networks revolves around discovering new routing algorithms which are able to model the connections between Capsule from a lower layer $i$ and a higher layer $j$, with these mainly being iterative in nature. These iterative algorithms come with the drawback that the already computationally and memory-intensive process of determining how much lower Capsules correspond to higher Capsules has to be repeated multiple times.

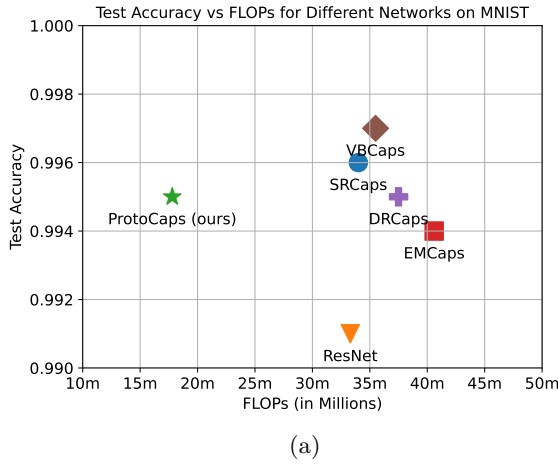 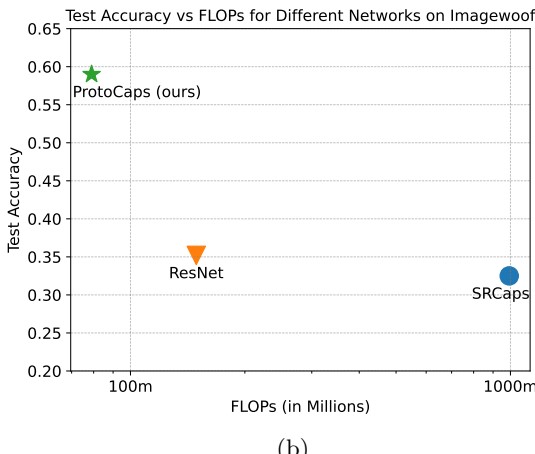

Figure 1: Comparative analysis of test accuracy vs Floating Point Operations (FLOPs, as calculated by the FVCore library (FAIR, 2023)) for various methods on two datasets. a) the MNIST dataset (LeCun et al., 2010), included for sanity to show our method is comparable in accuracy to other methods. b) the Imagewoof dataset (Howard, 2019), showing that we are able to outperform SR Caps and our ResNet18 baseline on a more difficult dataset in terms of image size. Unfortunately the iterative methods require too much gpu memory (>80GB) in order to process the Imagewoof dataset at comparable sizes. A big point to note is that while SRCaps scales by around 20 times in FLOP count when processing the larger feature maps of Imagewoof compared to MNIST, our model's increase is only about 4 times. An efficient method, in terms of low FLOPs and high performance, would be in the top left corner. Note the logarithmic scale for the x-axis in the Imagewoof plot.

We find that Capsule Networks with iterative routing mechanisms (e.g. Dynamic Routing (Sabour et al., 2017), Expectation Maximum Routing (Hinton et al., 2018), Routing Uncertainty in Capsule Networks (De Sousa Ribeiro et al., 2020) and Variational Bayes Routing (Ribeiro et al., 2020)) achieve the highest classification accuracy for Capsule Networks. We note that Self Routing Capsule Networks (SRCaps) method (Hahn et al., 2019) demonstrates strong performance with a significant improvement in speed due to the simple and non-iterative routing mechanism using routing sub-networks. Nonetheless, SRCaps still inherits the common drawback of high memory usage during the routing process due to the need to project and process a representation of each Capsule in $i$ to each Capsule in upper layer $j$, and any obvious extension to scale the complexity of its routing networks would nullify its speed advantage.

Motivated by these findings, we introduce *ProtoCaps*, a straight-through routing algorithm that offers lower memory usage based on trainable prototype soft clustering. Although our method underperforms compared to the slow iterative methods, our method substantially mitigates the memory consumption issue and provides an effective, efficient and scalable routing mechanism for Capsule Networks.

Our contributions can be summarised as follows:

- We propose a novel, non-iterative, trainable routing algorithm for Capsule Networks, setting a new benchmark for non-iterative routing methods.

- We validate the robustness of our ProtoCaps network across multiple standard datasets and introduce a more complex dataset with image characteristics akin to the ImageNet dataset (Deng et al., 2009).

- Our ablation studies reveal that while our routing algorithm shows robust performance, it can be further strengthened through additional architectural refinements.

## 2 Related Works

### 2.1 Capsule Networks

Capsule Networks are a different approach to neural architectures, introducing a hierarchical, part-to-whole relationship via computational entities known as Capsules. This unique structure facilitates the encapsulation of complex, multi-dimensional entities and relationships in the data, providing a more robust representation of the input space.

The fundamental building blocks of these networks are Capsules, which capture the pose (the state or attributes) of particular features in the input. Capsules $\mathbf{v}_i$ at a lower level predict their representation in higher-level Capsules $\hat{\mathbf{u}}_{j|i}$ via transformation matrices $\mathbf{W}_{ij}$, represented as follows:

$$\hat{\mathbf{u}}_{j|i} = \mathbf{W}_{ij}\mathbf{v}_i. \tag{1}$$

These predictions are not merely aggregated, but are rather routed to specific higher-level Capsules using routing coefficients depending on how much the lower-level Capsule determines it corresponds to the upper-level Capsule. These coefficients are calculated as:

$$C_{ij} = f(\hat{\mathbf{u}}_{j|i}, \mathbf{V}_j) \tag{2}$$

where $\mathbf{V}_j$ represents the higher-level Capsule, and $f$ is a routing function that determines the level of agreement between predictions and higher-level features. As these algorithms are typically iterative, $\mathbf{V}_j$ is initialised as if the lower Capsules are equally coupled.

The routed input, therefore, becomes a weighted sum of these predictions:

$$\mathbf{S}_j = \sum_i C_{ij}\hat{\mathbf{u}}_{j|i}. \tag{3}$$

Ultimately, each higher-level Capsule calculates its vector output from the routed inputs. This calculation can vary across different architectures, but is commonly represented as:

$$\mathbf{V}_j = g(\mathbf{S}_j) \tag{4}$$

where $g$ can represent a nonlinear activation function like ReLU, although some networks will employ custom activation functions.

This process of prediction, routing, and aggregation forms the core of Capsule Networks, allowing them to capture complex hierarchical relationships in the data, preserve spatial relationships, and provide robustness to transformations.

### 2.2 Routing Algorithms in Capsule Networks

The original Capsule Network architecture uses Dynamic Routing (Sabour et al., 2017) to iteratively refine inter-Capsule connections, introducing "coupling coefficients" to represent connection strength. These coefficients are updated using a softmax function over agreement values calculated as a dot product between a lower-level Capsule and a predicted output. After a fixed number of iterations, each higher-level Capsule's output is the weighted sum of lower-level Capsules' outputs.

Expectation-Maximization (EM) Routing (Hinton et al., 2018) is a mechanism that iteratively refines routing assignments and updates high-level Capsule activations, maximizing the likelihood of the input data. In the E-step, posterior probabilities of high-level Capsules are calculated. These are updated according to the current Capsule activations and the observed data. The M-step then updates the high-level Capsule

activations based on the current routing assignments and lower-level Capsules. The algorithm alternates between the E-step and M-step until convergence or for a fixed number of iterations.

Variational Bayes (VB) Routing (Ribeiro et al., 2020) addresses some challenges with EM Routing, including training instability and reproducibility. It applies Bayesian learning principles to Capsule Networks by placing priors and modelling uncertainty over Capsule parameters between layers. Initial routing coefficients are assigned, followed by the computation of votes for each Capsule. VB Routing then iterates over two key steps: updating the routing weights and the parameters. At the end of these iterations, VB Routing updates each Capsule's activation using a logistic function.

Self-Routing Capsule Networks (SR-CapsNet) (Hahn et al., 2019) confront the computational challenges of routing caused by iterative approaches by enabling each Capsule to independently calculate its routing coefficients using a small routing network. This eliminates the need for coordination between Capsules during the agreement process. In this model, each Capsule uses its routing network, taking inspiration from the concept of a mixture of experts (Masoudnia & Ebrahimpour, 2014), to directly estimate routing coefficients. This suggests that each Capsule specializes in different parts of the feature space, functioning in a manner akin to a collection of experts specializing in different regions of the input space.

SR-CapsNet computes routing coefficients ($C_{ij}$) and predictions ($\hat{\mathbf{u}}_{j|i}$) using two trainable weight matrices, namely $\mathbf{W}^{route}$ and $\mathbf{W}^{pose}$. Each of these matrices is equivalent to a fully connected layer for each Capsule in the higher layer. In every layer of the routing network, pose vectors ($\mathbf{u}_i$) are multiplied by the trainable weight matrix $\mathbf{W}^{route}$, resulting in the direct production of routing coefficients. Following softmax normalization, these coefficients are multiplied by the Capsule's activation scalar $a_i$ to generate weighted votes. The higher-layer Capsule's activation $a_j$ is then calculated as the summation of these weighted votes from lower-level Capsules across spatial dimensions ($H \times W$), or across $K \times K$ dimensions when using convolutions.

SR-CapsNet has demonstrated commendable performance on standard Capsule Network benchmarks. Nevertheless, this approach slightly hampers the Capsule Network's ability to dynamically adjust routing weights based on the input, as these weights are fully reliant on the learned parameters of the routing subnetworks.

Self attention mechanisms used in transformers Vaswani et al. (2017) have been applied to Capsule Network routing Shang et al. (2021); Duarte et al. (2021); Mazzia et al. (2021) by using $\hat{\mathbf{u}}_{j|i}$ as the query and key vectors to calculate attention scores and then using these as coupling coefficients for routing, with $\hat{\mathbf{u}}_{j|i}$ also acting as the value in the self attention mechanism.

## 2.3 Prototype Based Clustering

In both metric and unsupervised learning methods, prototype clustering (Caron et al., 2018) has emerged as an effective technique for categorizing image data without the requirement of manual labels. This methodology operates on the principle of extracting features from images by utilising a feature extraction network such as a CNN, subsequently clustering these features into distinct semantic groups via conventional algorithms, such as non-iterative k-means.

Following this, the model regards these cluster assignments as "pseudo-labels" for each image, and accordingly updates the feature extraction network to facilitate the prediction of the pseudo-label cluster for each image. This process, when performed iteratively through training, enables the CNN to progressively learn to generate feature representations that intrinsically group similar images into semantically significant clusters.

Prototype clustering thus serves as a pivotal technique for training models to find semantically meaningful clusters in an unsupervised fashion, effectively bypassing the need for manual labeling while ensuring meaningful image categorization.

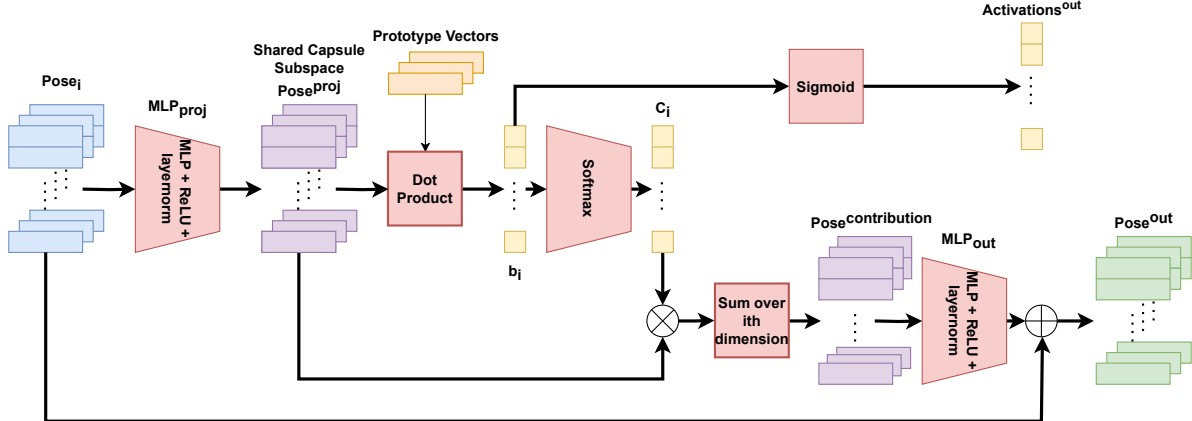

Figure 2: Diagram showing our proposed routing algorithm, i.e., ProtoCaps. Multiple layers of this ProtoCaps algorithm can be stacked in order to create a ProtoCaps Network. $\oplus$ and $\otimes$ denote elementwise addition and multiplication. We also include a residual connection from $Pose_i$ to the output of $\text{MLP}_{out}$ as discussed in section 3.3. Figure 4 shows the end to end ProtoCaps network.

## 3 Non Iterative Prototype Routing

### 3.1 Prototype Routing

Our routing algorithm uses a non-iterative approach to reduce computational overhead. In the following, we describe how the lower-level Capsules are mapped to the upper-level Capsules using our proposed prototype routing mechanism which is visualised in Figure 2. Each ProtoCaps routing layer functions by projecting the pose matrices of the lower-level Capsules into a shared subspace denoted as $S$. The projection is executed by using the following multi-layer perceptron ($\text{MLP}_{proj}$):

$$\text{pose}_i^{proj} = \text{MLP}_{proj}(pose_i), \tag{5}$$

where $pose_i$ denotes the $i$th pose matrix of the lower-level Capsules. Within the shared subspace $S$, there exists $n$ learnable prototype vectors $Q = (q_1, q_2, \cdots, q_n)$ where each $q_i \in R^{d \times 1}$. Here, $d$ signifies the dimensionality of the upper layer Capsules, and $n$ represents the number of Capsules in this upper layer, meaning that the number of Prototype vectors is equal to the number of Capsules in the next layer which we would like to route to. We initialise our Prototypes as trainable vectors from a normal distribution. Prototype vectors are updated in the backwards pass of training.

For each $i \in [1, m]$ where $m$ represents the number of Capsules in the lower layer, a new vector $b_i \in R^n$ is constructed, containing the unnormalised coupling coefficients. Each element of $b_i$ is the dot product of the pose of an $i$th lower-level Capsule and each of the $n$ prototype vectors:

$$b_i = (\text{pose}_i^{proj} \cdot q_1, \text{pose}_i^{proj} \cdot q_2, ..., \text{pose}_i^{proj} \cdot q_n), \tag{6}$$

which forms a coefficient matrix $B = (b_{ij}) \in R^{m \times n}$. In Figure 5 we show a simplified version of this process. We can see that the second image is closer in embedding space to prototype 1 so would be highly coupled to this prototype. The first image, on the other hand, is closest to prototype 2, but is still somewhat in the middle of the embedding space, and would therefore be routed more equally.

Subsequently, a softmax function is applied to each row of the $B$ to generate $C = (c_{ij})$, which ensures that the output vectors form a valid probability distribution representing how much each of the projected pose vectors corresponds to each prototype, i.e., $\sum_{j=1}^{n} c_{ij} = 1$.

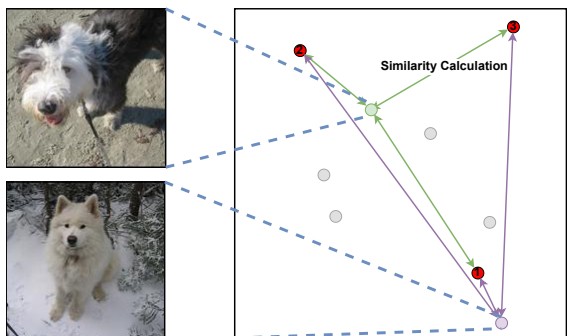

Figure 3: Diagram illustrating the intuition behind prototype routing. We have three prototypes denoted in red. Our pose matrix embeddings are shown in grey. For simplicity, we assume that an embedding represents an entire image and is derived directly from the image rather than the feature maps of the convolutional backbone. The dog images are sourced from the Imagewoof validation set (Howard, 2019).

Using these coupling coefficients $C$ we multiply the projected pose vectors $\text{pose}_i^{proj}$ to scale their contributions accordingly to the coupling coefficients,

$$\text{pose}_j^{contribution} = \sum_{i=1}^{m} c_{ij} \times \text{pose}_i^{proj} \tag{7}$$

where $j \in [1, n]$ and $i \in [1, m]$. These poses $\text{pose}_j^{contribution}$ are then fed through another MLP to provide the final poses for the Capsule layer, which will become the input poses for the following layer:

$$pose_j^{out} = \text{MLP}_{out}(\text{pose}_j^{contribution}). \tag{8}$$

Finally, we produce our activations for each Capsule by using the sigmoid function on our similarity vectors $b_i$, i.e,

$$\text{activations}_i^{out} = \text{sigmoid}(b_i). \tag{9}$$

A visual representation of this algorithm can be found in Figure 2.

### 3.2  Shared Capsule Subspace

In all of the Capsule Networks which we compare against in this paper, a common theme is to transform all pose vectors for Capsules from a previous layer $i$ into intermediate vote vectors $V_{ij}$ for each Capsule in a higher layer $j$. This generates a total of $i \times j \times h \times w$ vote vectors, where $h$ and $w$ represent the dimensions of the feature map, thereby leading to a high memory footprint for routing. This memory-intense process often restricts the scalability of these networks due to gpu memory limitations.

Our proposed method, however, takes a new approach. Instead of individually projecting each lower-level pose vector, we project them once into a shared feature space, where the prototype clustering takes place. This results in a reduction in the total number of vectors needed for routing by a factor of $j$. This significant reduction contributes to improved performance and lessened memory requirements, making our method more efficient and scalable for various applications.

### 3.3  Residual Connections

We incorporate residual connections (He et al., 2016) into our algorithm. The use of these connections alleviates the issue of vanishing gradients, a common hurdle in training deep networks, including Capsule

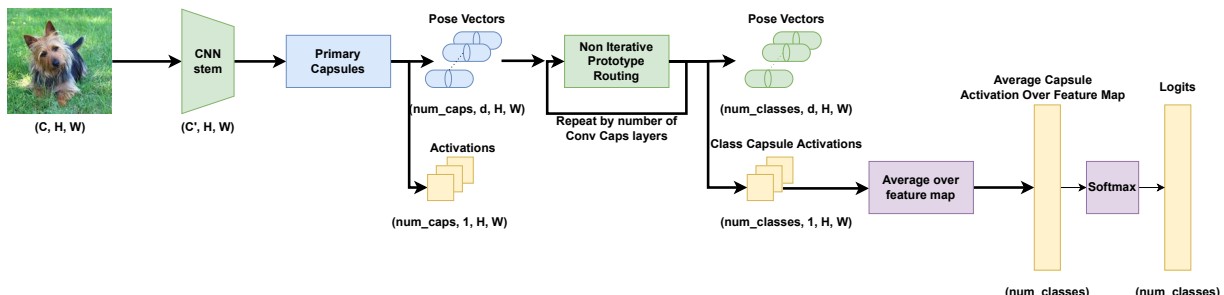

Figure 4: Diagram illustrating the end to end structure of our ProtoCaps network, including tensor dimensions for clarity. Note that the Class Capsule layer (the Capsule layer which corresponds to the classification prediction) is incorporated within the Non-Iterative Prototype Routing loop (i.e., ProtoCaps, see Figure 2). This functions in a similar manner to every other ProtoCaps layer except for its number of Capsules equates to the number of classes rather than being a hyperparameter of the network design. The dog image used is sourced from the Imagewoof validation set (Howard, 2019).

Networks (Mitterreiter et al., 2023). They allow the gradient to directly flow back through the layers, reducing the risk of it diminishing to the point where earlier layers struggle to learn.

Additionally, residual connections help combat the degradation problem, where adding more layers can inadvertently increase training error. By providing a shortcut for the information flow across layers, they allow the network to focus on learning the residual function, essentially the deviation from the identity mapping, making the learning process more efficient.

In ProtoCaps, residual connections play a key role in preserving valuable knowledge from earlier layers. They prevent this information from being lost through the scaling mechanism, maintaining the richness of initial data and promoting smoother and more effective training. This results in a more robust and performant network.

## 4 Experiments

In this section we will focus on first comparing our routing algorithm against the state of the art in Capsule Networks in terms of both efficiency and classification accuracy, additionally, we also compare against a ResNet-18 from the Timm library (Wightman, 2019) as a baseline. We then show ablation studies on our network, showing the performance change when adding residual connections. For a fair comparison with our most closely related work, SRCaps (Hahn et al., 2019), we use both the small CNN and ResNet backbones proposed in their paper.

### 4.1 Experimental Design

#### 4.1.1 Datasets

In this study, we conduct benchmark tests on five diverse datasets, with most being standard in Capsule Network research. We commence our evaluations with the MNIST dataset (LeCun et al., 2010), considered the foundational dataset for image classification. This initial phase allows us to ascertain our network's base performance, prior to engaging in more computationally demanding tests. Subsequently, we employ the more challenging FashionMNIST (Xiao et al., 2017) and CIFAR10 datasets (Krizhevsky et al., 2009) to test our network's proficiency in classifying more complex images. For these datasets, we do not use any augmentations other than a random resized crop of 0.8 to 1.0 of the image during training.

To further assess the capabilities of our network in terms of generalizing to unseen viewpoints which is a characteristic trait of Capsule Networks, we utilize the SmallNORB dataset (LeCun et al., 2004). For SmallNORB, we employ two transformations to improve our network's learning: Random Resized Crop and Color Jitter. Random Resized Crop enhances the model's translation invariance by altering the framing

Table 1: The results for various models on five data sets. FLOP count refers to the amount of floating point operations performed in a single forward pass of the network for a single Cifar10 image Krizhevsky et al. (2009) and is given as an approximate measure of the computational complexity of each network. All networks are trained using official implementations or open source implementations which achieve the reported results of papers where possible. Unfortunately, it was computationally infeasible to train DR, EM and VB Caps on the Imagewoof dataset due to memory requirements exceeding maximum GPU memory available.

| | FLOP Count (Millions) | MNIST | Fashion MNIST | Cifar10 | SmallNORB | Imagewoof |
|---|---|---|---|---|---|---|
| ResNet18 (He et al., 2016) | 74.5 | 99.1 | 89.3 | 78.4 | 89.8 | 52.7 |
| MobileNetv3 (Howard et al., 2019) | 17.6 | 99.5 | 98.7 | 77.1 | 83.3 | 45.9 |
| ViT-S (Dosovitskiy et al., 2021) | 810 | 99.4 | 99.3 | 80.0 | 91.1 | 57.3 |
| DR Caps (Sabour et al., 2017) | 73.5 | 99.5 | 82.5 | 91.4 | 97.3 | - |
| EM Caps (Hinton et al., 2018) | 76.6 | 99.4 | - | 87.5 | - | - |
| VB Caps (Ribeiro et al., 2020) | 70.8 | 99.7 | 94.8 | 88.9 | 98.5 | - |
| SR Caps (Hahn et al., 2019) | 62.2 | 99.6 | 91.5 | 82.7 | 92 | 32.5 |
| ProtoCaps (ours) | 32.4 | 99.5 | 92.5 | 87.1 | 94.4 | 59.0 |

of the image's subject. Colour Jitter modifies brightness and contrast, expanding the dataset to simulate various lighting conditions. Both transformations serve to augment the data, strengthening the network's generalization ability. Lastly, to examine our network's ability to process highly intricate images, we make use of the Imagewoof dataset (Howard, 2019). As a subset of ImageNet (Deng et al., 2009) featuring 10 dog breeds at 224x224 image resolution, this data set poses a unique challenge since Capsule Networks typically struggle with such demanding images due to high computational requirements. For this particular dataset, we conducted experiments with various image sizes, ultimately finding that a 64x64 resolution produced the best results. Additionally, we only incorporate a random resized crop technique that scales the image within a range of 0.8 to 1.0. This approach enables us to further optimize our network's performance while handling these complex images.

### 4.1.2 Architecture

A typical Capsule Network architecture encompasses a convolutional stem composed of one or more convolutional layers, a Primary Capsule layer that translates the feature map procured by the convolutional stem into a Capsule structure, Class Capsules - the terminal layer of the network, the activations of which are utilized as logits for softmax classification, and optionally, one or more intermediate convolutional Capsule layers that reside between the primary and class Capsules to expand the depth of the parse tree.

For our research, we adopt the convolutional stems directly from SRCaps (Hahn et al., 2019) for a comprehensive comparison. Depending on the context, we employ either a ResNet20, comprising 19 convolutional layers with the terminal fully connected layer replaced by our Capsule Network, or a simpler convolutional network of 7 layers devoid of residual connections. As for the Primary, Convolutional, and Class Capsules, we retain the existing structure but substitute the routing algorithm with our innovative straight-through Prototype routing. An illustration of our architecture can be seen in Figure 4.

### 4.1.3 Training Strategy

To train our Networks, we use the cross entropy loss function and Adam optimizer with weight decay. We train for 350 epochs and use a fixed learning rate scheduler which decreases the learning rate at 150 and 250 epochs and a batch size of 64. This is the same as in the SRCaps paper (Hahn et al., 2019). For SmallNorb, we instead train for 100 epochs to avoid overfitting on this simple dataset. For comparing those algorithms in terms of computational complexity, we use Floating Point Operation (FLOP) counts which refers to the amount of floating point operations performed in a single forward pass of the network. This metric is calculated using the FVCore library (FAIR, 2023) with a batch size of 1 from the target dataset.

Table 2: Full ablation results on all tested datasets, showing the test accuracy results of adding residual connections to our network. The backbone used for these experiments is the ConvNet variant.

| | Without Residual Connections | With Residual Connections |
|---|---|---|
| MNIST | **99.4** | **99.4** |
| FashionMNIST | 91.1 | **92.5** |
| SmallNORB | 92.2 | **94.4** |
| Cifar10 | 75.9 | **85.5** |
| Imagewoof | 38.1 | **59.0** |

Table 3: Full ablation results on all tested datasets, showing the difference in test accuracy of our ProtoCaps model using the ConvNet and ResNet backbones proposed in SRCaps (Hahn et al., 2019).

| | ConvNet Backbone | ResNet Backbone |
|---|---|---|
| MNIST | **99.5** | **99.5** |
| FashionMNIST | **92.5** | 92.1 |
| SmallNORB | **94.4** | 92.6 |
| Cifar10 | 85.5 | **87.1** |
| Imagewoof | 56.4 | **59.0** |

### 4.2 Results

#### 4.2.1 Results of Image Classification

Table 1 presents the classification results of our study, where we juxtapose four major Capsule architectures alongside a ResNet18 (He et al., 2016), approximately matching in computational complexity to our Capsule Networks. While our method may not surpass the iterative approaches such as DR, EM, and VB Caps in classification accuracy, it stands out for its significantly lower requirement of FLOPs and memory, thus showcasing its desirable properties.

Compared to non-iterative models like SR Caps and ResNet18, our proposed approach either aligns with or outperforms their results, all the while maintaining superior computational efficiency. We are beaten in terms of efficiency by the MobileNetv3, but perform better in model performance. We are beaten or comparable in standard image classification tasks by the vision transformer, but perform better on novel viewpoint generalisation as shown by the SmallNORB results. This reinforces the efficacy of our method, demonstrating its potential to deliver accurate results without imposing heavy computational demands. This would be a crucial step towards making Capsule Networks competitive as image classifiers in the real world.

### 4.3 Ablation Studies

In this section we will show the results of the modification of certain elements within our network, showing how we were able to converge on our best model showcased in Table 1.

#### 4.3.1 Residual Connections

Table 2 shows how our network performs with and without residual connections to show that residual connections contribute to the strong accuracy that we are able to provide, rather than being solely responsible. In particular, these results show that while all datasets benefited from the network containing residual connections, it was the more difficult datasets which benefited more from the residual connections, likely due to their higher complexity.

Table 4: Results of going deeper with ProtoCaps. We include the Floating Point Operations (FLOPs) of different depth (number of Convolutional Capsules layers) and test accuracy on our five datasets. All results are using the Convolutional backbone from Table 3 to show how the network scales with the size of the dataset.

| | FLOPs | 1 | FLOPs | 2 | FLOPs | 3 | FLOPs | 4 | FLOPs | 5 | FLOPs | 10 |
|---|---|---|---|---|---|---|---|---|---|---|---|---|
| MNIST | 17.1m | 99.4 | 17.4m | **99.5** | 17.8m | **99.5** | 18.1m | 99.4 | 18.3m | **99.5** | 19.8m | **99.5** |
| FashionMNIST | 17.1m | 90.9 | 17.4m | **91.7** | 17.8m | 91.3 | 18.1m | 91.4 | 18.3m | 91.6 | 19.8m | 91.6 |
| SmallNORB | 18.7m | 91.3 | 19.0m | 93.5 | 19.3m | **94.4** | 19.6m | 93.7 | 19.9m | 90.5 | 21.4m | 92.8 |
| Cifar10 | 37.1m | 82.7 | 19.4m | 82.7 | 19.7m | **85.5** | 20.0m | 84.0 | 20.3m | 83.9 | 21.8m | 84.5 |
| Imagewoof | 76.6m | 52.8 | 77.8m | 56.1 | 79.0m | **56.5** | 80.2m | 52.6 | 81.4m | 53.1 | 87.5m | 54.3 |

### 4.3.2 Convolutional vs ResNet backbone

In our experimental design, we referenced the dual approach adopted by SRCaps (Hahn et al., 2019), wherein both a ResNet and a concise convolutional network were employed interchangeably to achieve optimal results for different datasets. SRCaps posited that such a configuration was a necessity, attributing the improved performance on simpler datasets to the use of a less powerful feature detector in order to alleviate overfitting.

In this section, we compare the impact of both network backbones on our ProtoCaps model. Our findings, as depicted in Table 3, indicate a similar trend, where simpler datasets indeed favor a less complex backbone, potentially due to the same overfitting concerns. However, the performance enhancement derived from utilizing the ResNet-like backbone is not overwhelmingly significant. Consequently, in scenarios where fast inference is of importance, the adoption of the ConvNet backbone might be a reasonable compromise across all use cases, as it does not incur substantial losses in performance.

### 4.3.3 Going Deeper with ProtoCaps

The examination of scalability in the context of our proposed ProtoCaps network forms a significant part of this discussion, while we have shown that smaller networks work well, the established literature indicates that model capacity is important for the hardest datasets.

By systematically varying the quantity of convolutional Capsule layers implemented within the ProtoCaps networks, we are able to conduct a comprehensive analysis across five distinct datasets. We report on the consequential variation in FLOPs, alongside its subsequent impact on test accuracy in Table 4.

Contrary to the observed performance improvements commonly associated with the introduction of residual connections in Convolutional Neural Networks (CNNs) (He et al., 2016), the integration of these connections within our ProtoCaps network primarily serves to preserve network stability. Unfortunately, they do not contribute significantly to enhancing the network's performance with increased scaling. Instead, the network saturates around 2-3 layers and then no longer improves, in contrast to other Capsule Networks which tend to start to devolve as larger amounts of layers are added (Everett et al., 2023). This distinct behavior necessitates further investigation and offers a compelling area for future exploration.

## 5 Conclusion

In this paper, we've introduced *ProtoCaps*, a novel, trainable and non-iterative routing algorithm for Capsule Networks that significantly enhances processing power and memory efficiency. Our extensive experiments and detailed ablation studies highlight the effectiveness of the different components within our proposed model. Furthermore, by benchmarking our method against a challenging, real-world ImageNet (Deng et al., 2009) size dataset, we've brought Capsule Networks a step closer to effectively handling modern data sets.

Looking ahead, we're eager to delve deeper into replicating the scaling laws observed in transformers and CNNs within Capsule Networks. Specifically investigating the properties of the Shared Capsule Subspace to determine whether better initialisation of Prototypes would help. This line of investigation aims to unearth

potential benefits from enhancing the depth of our proposed models, further expanding the capabilities and applications of Capsule Networks.

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

# A    Appendix

## A.1    Ablation of Number of Capsules

Table 5: Ablation study to show our choice of number of Capsules per layer. 16 and 32 are standard values to Capsule Networks, but we additionally experiment with 64 and 128 Capsules per layer.

|              | 16   | 32   | 64   | 128  |
|--------------|------|------|------|------|
| MNIST        | **99.5** | **99.5** | **99.5** | **99.5** |
| FashionMNIST | 92.2 | **92.5** | 92.4 | **92.5** |
| Cifar10      | 86.7 | **87.1** | **87.1** | 87.0 |
| SmallNORB    | 93.2 | **94.4** | 94.3 | **94.4** |
| Imagewoof    | 53.8 | **59.0** | 58.9 | **59.0** |

In table 5 we show ablation studies of our highest performing network, as shown in table 1. This table shows that while 16 Capsules per layer was not enough to maximise the networks performance, going beyond 32 Capsules did not provide a benefit but would increase the computational demand. Thus our final architecture is comprised of 32 Capsules per layer. Why the network did not see improvement from a wider layer is currently unknown and is an avenue of research that can be investigated in the future.

## A.2 Capsule Visualisations

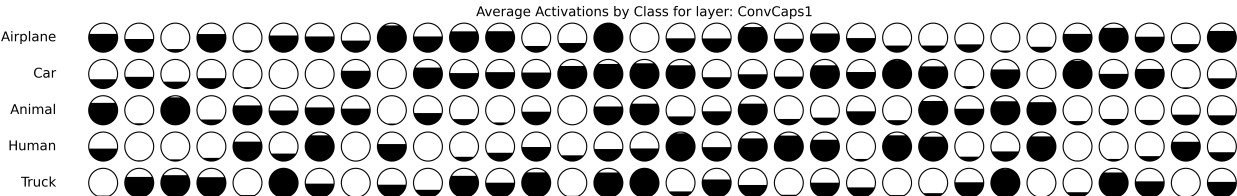

Figure 5: Average activations of each Capsule unit in the Convolutional Capsules (ConvCaps) layer of a 3 layer Capsule Network (Primary Caps, Conv Caps, Class Caps) trained on the smallNORB dataset LeCun et al. (2004). The amount that each circle is filled shows how much over the entire test set for each class the Capsule is activated on average e.g. a circle with no fill represents a Capsule which was never activated for this class, while a circle with full fill represents a Capsule which is always activated. Each row shows a different class for easy comparison.

In figure 5 we visualise the average capsule activation per class of the Convolutional Capsule layer of a three layer Capsule Network trained on the smallNORB dataset LeCun et al. (2004). This image shows that certain Capsules are specialising to certain classes, in particular it is interesting to note that Capsules which represent classes with some overlapping features, e.g. truck, airplane and car share some commonly highly activated Capsules. We only show the ConvCaps layer as the ClassCaps layer will trivially always have the Capsule corresponding to the correct class as the dominant activated Capsule and the PrimaryCaps layer has been through any routing mechanism.

