# OpenReview forum: "ProtoCaps: A Fast and Non-Iterative Capsule Network Routing Method"
_TMLR — Accepted by TMLR_

### Review · Reviewer_kxXW · 2023-10-04

**Summary Of Contributions:**

This work introduces ProtoCaps, a routing algorithm for Capsule Networks. ProtoCaps offers a non-iterative approach inspired by trainable prototype clustering, reducing memory usage during training. It sets a new benchmark for non-iterative routing methods, demonstrating robustness across multiple datasets, including a complex one akin to ImageNet. The paper's ablation studies reveal room for further improvement through architectural refinements. By benchmarking against a challenging dataset, it brings Capsule Networks closer to effectively handling modern and complex data, while future research directions aim to enhance the model's capabilities and applications.

**Audience:**

Yes

**Claims And Evidence:**

No

**Requested Changes:**

Please address the points in my weakness section.
I am listing a few critical adjustments that would strengthen the work but also increase the likelihood of a positive score.

**Critical Adjustments:**
- Proper performance comparison: Comparing to SOTA methods and best-effort baseline methods.
- Comparison to other light models.
- Convincing evidence that capsule networks have the potential to solve a problem that SOTA networks can't.

**Strengths And Weaknesses:**

## Strengths
- The proposed ProtoCaps is technically novel
- The improvement in terms of the FLOP Count of of ProtoCaps is significant compared to other Capsule Network architectures.
- The paper is well-written and easy to follow

## Weakness
- The result comparison is insufficient:
	- The compared performances seem unreasonable. Especially the performance of ResNet. For example, various resources exist online, which show a higher performance of ResNet18 (e.g. CIFAR10 [1])
	- The performance of ResNet18 on ImageWoof seems unreasonable. On the full ImageNet dataset, ResNet18 achieves 69.758%, hence I am expecting a performance ideally higher or on par with that. Here [2] a user reported "75% accuracy without any bells and whistles".
	- These days transformer architectures emerge as the strongest architectures in computer vision. A comparison to these is entirely missing in the paper.
	- Since this work attempts to argue for the speed benefits of ProtoCaps, a comparison to other "light" models is missing, such as MobileNet, EfficientNet or mobile Transformer works.
- Considering the strong limitations of Capsule Networks, it is questionable if this work is of interest to the research community since much stronger networks exist. The authors fail to provide evidence that SOTA methods fail to solve tasks that Capsule Networks might be able to solve. The only evidence for this is shown in a toy example with RN18 on a relatively small toy dataset (SmallNORB).

[1] https://github.com/kuangliu/pytorch-cifar
[2] https://github.com/fastai/imagenette/issues/32

---

> ### Author Response · Authors · 2023-11-08
> **Review response to kxXW**
>
> We would like to take this opportunity to thank you for your review of our paper and hope that we can alleviate your concerns below.
>
> ``` The compared performances seem unreasonable. Especially the performance of ResNet. For example, various resources exist online, which show a higher performance of ResNet18 (e.g. CIFAR10 [1]) ```
>
> The ResNet18 results of our network may seem weak compared to established baselines. But it is important to ensure that they were trained with the same data augmentations as our Capsule Networks for fair comparisons.
>
> Regarding the GitHub repository you have linked for comparison. We have found issues with their experimental setup which coupled with the addition of augmentation strategies improves their results:
>
> https://github.com/kuangliu/pytorch-cifar/issues/136 - The ResNet-18 implementation that they use differs from the original and contains approx. 30% more parameters.
>
> https://github.com/kuangliu/pytorch-cifar/issues/145 - They do not split their training set into a train/validation set as is standard and was the methodology in our experimental results. Instead they use the test set to evaluate the network after each training epoch and select the best model epoch checkpoint based on this accuracy, meaning that parameters effected by validation accuracy not improving, such as weight decay and learning rate values are guided by data which should be completely unseen. They also receive the advantage of more training data as they are removing samples from the training set for the validation set.
>
> We have re-run our ResNet-18 results for all datasets using the huggingface pytorch-image-models official training script [1], taken the highest result and updated the paper accordingly. Although this does not improve for anything except the imagewoof results. However, we do note that performance is improved when using the full augmentation suite that the repository leverages.
>
> For full transparency and reproducibility the exact command we use for launching the training script is:
>
> python3 train.py torch/cifar10 –model resnet18 –dataset-download –data-dir $DATA_DIR –dataset torch/cifar10 –no-aug –img-size 32
>
> ``` The performance of ResNet18 on ImageWoof seems unreasonable. On the full ImageNet dataset, ResNet18 achieves 69.758%, hence I am expecting a performance ideally higher or on par with that. Here [2] a user reported "75% accuracy without any bells and whistles". ```
>
> Unfortunately the user who submitted the github issue referenced hasn’t provided any proof of these experiments despite being asked by the repo maintainer. As mentioned above we ensured that our results were not unfairly evaluating the ResNet and followed instructions at [2] for training pytorch-image-models [1] models on imagenette, which is also a subset of imagenet similar to imagewoof, but replaced imagenette with imagewoof. This increased the accuracy of the resnet18 and the paper has been updated accordingly. Once again, the training command we use is listed below.
>
> python3 train.py $DATA_DIR/imagewoof –model resnet18 –no-aug
>
> ``` These days transformer architectures emerge as the strongest architectures in computer vision. A comparison to these is entirely missing in the paper. ```
>
> We have added comparisons to vision transformers to table 1. We evaluated ViT Small, TinyViT and MobileViT which all have similar parameter counts to our network, but only include the results of the consistently best performing architecture which was ViT Small in order to not overclutter our results table.
>
> ``` Since this work attempts to argue for the speed benefits of ProtoCaps, a comparison to other "light" models is missing, such as MobileNet, EfficientNet or mobile Transformer works. ```
>
> We have evaluated MobileNetv3_large_100 and EfficientNet_b0 from the pytorch-image-models library [1], these specific models are the closest in terms of parameter count to our Capsule Network. We found that MobileNetv3_large_100 consistently performed the best and have added the results to table 1.
>
> ``` Considering the strong limitations of Capsule Networks, it is questionable if this work is of interest to the research community since much stronger networks exist. The authors fail to provide evidence that SOTA methods fail to solve tasks that Capsule Networks might be able to solve. The only evidence for this is shown in a toy example with RN18 on a relatively small toy dataset (SmallNORB). ```
>
> We have added evaluations on the SmallNORB dataset for other SOTA methods to show that they also fail to generalise to novel viewpoints as effectively as Capsule Networks can. We believe this gives Capsule Networks a clear purpose for investigation in the machine learning landscape.
>
> [1] https://github.com/huggingface/pytorch-image-models
>
> [2] https://timm.fast.ai/training_scripts

---

### Review · Reviewer_2aMV · 2023-10-07

**Summary Of Contributions:**

This paper proposes a trainable and non-iterative routing algorithm for Capsule Networks that significantly enhances processing power and memory efficiency. With the attention-like routing architecture, the authors realize straight-through algorithm and the increased FLOPs is fewer compared to previous methods when processing large features. The experiments underscores the potential of proposed methodology in enhancing the operational efficiency and performance of Capsule Networks.

**Audience:**

Yes

**Broader Impact Concerns:**

No.

**Claims And Evidence:**

Yes

**Requested Changes:**

1. In sec3.1, the Convolutional Capsules layer contains a learnable prototype vectors, what is the meaning of this scheme? When there are multiple capsule layers, how to set the learnable prototypes?
2. Through the settings and methods in the paper, I cannot figure out the computation comparison between the resnet and this work.
3. In Figure 2, what is the meaning of the bottom residual connection? Besides, the meaning of sign $\oplus$ is not clear.

**Strengths And Weaknesses:**

Strengths:
1. By designing a straight-through algorithm, the proposed method can alleviate the large computation caused by iterative routing method.
2. In the research of capsule network, the routing strategy is very important. This paper constructs a shared capsule subspace which can save memory I/O greatly.

Weakness:
1. As far, the capsule network has been researched several years, but its scalability and practicality are still questionable. Besides, as the development of vision transformer, the patches or learnable queries in vision features are exactly similar to the capsule in Capsule network.
2. The training in large scale dataset is still limited for Capsule network, and it is expected.
3. The novelty is somewhat weak, the computation of coupling coefficients and output vector pose is very similar to the popular self-attention mechianism. There are many capsule network has already combined with self-attention[1,2,3, etc], its better to clarify there disfferances.
[1] Efficient-CapsNet: capsule network with self-attention routing
[2] Routing with Self-Attention for Multimodal Capsule Networks
[3] Capsule Network Based on Self-Attention Mechanism

---

> ### Author Response · Authors · 2023-11-08
> **Review response to 2aMV 1/2**
>
> Firstly, thank you very much for your review. We very much appreciate the constructive criticisms provided and hope that we can alleviate the issues which you had with our paper.
>
> ```As far, the capsule network has been researched several years, but its scalability and practicality are still questionable. Besides, as the development of vision transformer, the patches or learnable queries in vision features are exactly similar to the capsule in Capsule network. ```
>
> Scalability is a property that is desirable for any algorithm, and while we haven’t quite managed to find a model that is scalable to the extent of Vision Transformers, ConvNexts etc, we believe that our work is a step in the right direction by providing a Capsule Network which, while it does not improve significantly from scaling, can be reasonably scaled computationally without the unjustifiable levels of compute/VRAM required by other top performing Capsule architectures.
>
> Our routing differs from vision transformers in that we do not create our prototype vectors (which could be seen as similar to queries in self attention) by a projection of the data, but they are learnable parameters of the model instead.
>
> Finally, patch based encodings are becoming popular across all architectures of computer vision, with vision transformers [3], convnexts [4] and conv-mixers [5] all using patch embeddings as the first step of their architectures.
>
> ``` The training in large scale dataset is still limited for Capsule network, and it is expected. ```
>
> While the results are limited on the large scale dataset compared to modern vision transformers, being able to process images of this size/complexity is something that a lot of Capsule Network architectures are unable to do. We hope that our work will encourage others to test their architectures on harder datasets rather than just datasets where Capsules are known to perform well.
>
> Additionally, we hope that our work is a step towards a stackable Capsule Network which share the scaling properties of vision transformers and Convolutional Networks (ConvNets).
>
> ``` The novelty is somewhat weak, the computation of coupling coefficients and output vector pose is very similar to the popular self-attention mechianism. There are many capsule network has already combined with self-attention[1,2,3, etc], its better to clarify there disfferances. ```
>
> Similarly to our response to your first weakness, we believe that our method differs enough from the self attention mechanism and the routing algorithms proposed in the papers which you have referenced. While these papers leverage the self attention mechanism, or a variant of it, our method differs in that we do not generate our $\text{Pose}_{proj}$ and prototype vectors $Q$ from the input data. Our $Q$ vectors are learnable parameters of the model which through training learn to route the representations from lower layers of Capsules to the upper layer of Capsules. We have added comparisons to the papers requested briefly in our related works. For an in depth comparison on the similarities between the routing procedure in Capsule Networks and the Self Attention Mechanism from transformers, please see section 5 in [1].
>
> We will now address the changes which we have made to our manuscript in order to comply with your requested changes
>
> ``` In sec3.1, the Convolutional Capsules layer contains a learnable prototype vectors, what is the meaning of this scheme? When there are multiple capsule layers, how to set the learnable prototypes? ```
>
> We have updated section 3.1 to include more details on what the number of prototypes represents, how they are trainable parameters of the model and how we instantiate them.
>
> ``` Through the settings and methods in the paper, I cannot figure out the computation comparison between the resnet and this work ```
>
> The comparisons between the resnet, other non-capsule architectures and our work is quantified by the amount of FLOPs (FLoating Point Operations). We believe that this is a reasonable metric to compare architectures as it quantifies the raw amount of computations which must be completed per image. We do not use throughput or training time as a metric as deep learning libraries contain many optimizations specifically to increase the rate at which operations for Vision Transformers and ConvNets can improve their FLOPs per second. If Capsule Networks become popular in the same way which these architectures are, efforts would be made by the community to perform the same low level optimizations for Capsule Networks.
>
> Whenever we discuss FLOPs in our paper we have ensured that it is clear that this is calculated by the FVCore library [2] for all networks to ensure fair comparison.

---

> ### Author Response · Authors · 2023-11-08
> **Review response to 2aMV 2/2**
>
> ``` In Figure 2, what is the meaning of the bottom residual connection? Besides, the meaning of sign ⊕ is not clear. ```
>
> We agree that this was unclear. We have adjusted the caption to clearly show that the residual connection performs elementwise addition with the output of $\text{MLP}_{out}$ and pointed towards section 3.3 for justification about why we do this. We have also clarified the meaning of all symbols which we have used.
>
> [1] Learning with capsules: A survey - https://arxiv.org/pdf/2206.02664.pdf
>
> [2] https://github.com/facebookresearch/fvcore/blob/main/docs/flop_count.md
>
> [3] An Image is Worth 16x16 Words: Transformers for Image Recognition at Scale - https://arxiv.org/pdf/2010.11929.pdf
>
> [4] ConvNeXt V2: Co-designing and Scaling ConvNets with Masked Autoencoders - https://arxiv.org/pdf/2301.00808.pdf
>
> [5] Patches Are All You Need? - https://arxiv.org/pdf/2201.09792.pdf

---

### Review · Reviewer_UVna · 2023-11-01

**Summary Of Contributions:**

CapsNets are a new type of deep learning architecture, which is slow and expensive to train. This paper introduces ProtoCaps, a new way to train CapsNets that is faster and cheaper without sacrificing performance. Specifically, it uses a straight-through routing algorithm that offers lower memory usage based on trainable prototype soft clustering.
The authors also show that their new approach works well on a large and challenging dataset.

**Audience:**

Yes

**Claims And Evidence:**

Yes

**Requested Changes:**

Please address the following questions:

+ How to decide the number of prototype vectors? I do not see any discussions and experiments to investigate how the number of prototype vectors affects the model performance and how to choose the number.

+ How to learn the prototype vectors? The authors did not provide the details on how to learn these vectors. How to ensure the model can learn expected prototype vectors.

+ Are the prototype vectors embeddings for each category in the datasets? I suggest the authors provide some visualization to interpret the leaned prototype vectors.

+ For results, why does the proposed method achieve lower performance on the SmallNORB compared to VB Caps and DR Caps?

**Strengths And Weaknesses:**

Pros:

+ The authors propose a novel non-iterative, trainable routing algorithm for Capsule Networks, which is different from existing iterative methods and is technically sound.

+ The proposed approach is shown to be effective and efficient on several datasets, including MNIST, Fashion-MNIST, Cifar10, SmallNORB, and Imagewoof.



Cons:

The prototype is the key to the proposed new routing method. I have several concerns and questions about it.

+ How to decide the number of prototype vectors? I do not see any discussions and experiments to investigate how the number of prototype vectors affects the model performance and how to choose the number.

+ How to learn the prototype vectors? The authors did not provide the details on how to learn these vectors. How to ensure the model can learn expected prototype vectors.

+ Are the prototype vectors embeddings for each category in the datasets? I suggest the authors provide some visualization to interpret the leaned prototype vectors.

+ For results, why does the proposed method achieve lower performance on the SmallNORB compared to VB Caps and DR Caps?

---

> ### Author Response · Authors · 2023-11-08
> **Review response to UVna**
>
> Firstly, as with the other reviewers we thank you for your feedback on our work. We have made changes in order to improve the paper based upon your requested changes.
>
> ```How to decide the number of prototype vectors? I do not see any discussions and experiments to investigate how the number of prototype vectors affects the model performance and how to choose the number.```
>
> The number of prototype vectors is equal to the amount of Capsules in the upper layer. The number which is chosen is a hyperparameter of the model similar to the number of feature maps a convolutional layer produces. We have followed conventional knowledge from other papers and chosen 32 as the number of Capsules per layer, but have also experimented with 16, 64 and 128 per layer, with 32 ultimately performing the best due to marginal, if at all, gains from adding larger amounts of Capsules per layer and 16 Capsules underperforming. These results have been added to the paper in table 5 of the appendix.
>
> We believe that there is room in the future for papers to investigate how the number of Capsules per layer can be increased, akin to papers such as the Wide Resnet [1] as this is another scaling factor of the network.
>
> ```How to learn the prototype vectors? The authors did not provide the details on how to learn these vectors. How to ensure the model can learn expected prototype vectors.```
>
> We have updated our methodology section to make it clear that these prototype vectors are learnt and updated via backpropagation, at the same time as every other parameter of the model. As for how to ensure that the model learns the correct prototype vectors, we view this similarly to how a convolutional neural network (CNNs) does not provide any hard mechanism to force kernels to learn specific features, but rather that it will learn internal representations which are able to minimise the loss function and therefore maximise the accuracy of the model.
>
> ```Are the prototype vectors embeddings for each category in the datasets? I suggest the authors provide some visualization to interpret the leaned prototype vectors.```
>
> As stated above, we do not enforce the representations learnt by the prototypes to be anything specifically in order to allow the model to learn the prototypes organically. In order to visualise whether prototypes learn class specific features, we have added to our appendix a visualisation of which capsules are highly active in each layer for the test set per class. These show that each class has, on average, capsules which do and do not activate, showing specialisation of certain capsules to concepts of each class.
>
> ```For results, why does the proposed method achieve lower performance on the SmallNORB compared to VB Caps and DR Caps?```
>
> It is currently an open question as to why routing works, with recent works proposing that routing in Capsule Networks might not be building a dynamic parse tree as we assumed [2].
>
> As to why these routing methods outperform our own, it is likely due to all routing coefficients being calculated directly from the current minibatch through iterative methods. However, these methods are extremely slow due to the number of calculations which need to be made, along with the loops when compared to vision transformers, CNNs and straight through routing mechanisms. Thus our main contribution here is to provide a Capsule Network routing algorithm which can be efficiently stacked without using unreasonable amounts of computational power but minimally compromise on performance.
>
> [1] Wide Residual Networks - https://arxiv.org/pdf/1605.07146.pdf
>
> [2] Why Capsule Neural Networks Do Not Scale: Challenging the Dynamic Parse-Tree Assumption - https://arxiv.org/pdf/2301.01583.pdf

---

> > ### Comment · Reviewer_UVna · 2023-11-14
> > **Response to authors**
> >
> > Thank the authors for responding to my questions! My main concerns have been satisfactorily resolved.

---

### Decision · Action_Editor_qYhQ · 2023-12-05

**Recommendation:** Accept as is

**Comment:**

Some minor format change requests:

- Please make sure that Table 1 fits horizontally within the paper margins
- Please increase all fonts in Figure 1 (ticks, labels etc) for better readability. Same for the other Figures, if possible.

**Audience:**

I believe that researchers working on Capsule Networks will find the paper's method and findings interesting. As reviewers also point out, this direction still has many problems need to be solved but the paper provides some advances in this direction.

**Claims And Evidence:**

The paper makes a number of claims/contributions and proposes a new routing method for capsule networks. Claims are sufficiently supported.

---

> ### Author Response · Authors · 2023-12-06
> **Camera ready paper submitted**
>
> Hello
>
> Firstly we would like to thank all of the reviewers along with yourself for the smooth process of submitting to TMLR.
>
> We have uploaded the final copy of our paper taking into account your format change requests and have added a link to the repository which will contain the code.
>
> Thanks